# Lifecourse investigation of the cumulative impact of adversity on cognitive function in old age and the mediating role of mental health: longitudinal birth cohort study

Yiwen Liu ![ORCID],[1] Praveetha Patalay ![ORCID],[1,2] Jean Stafford,[1] Jonathan M Schott,[3] Marcus Richards[1]

[1]MRC Unit for Lifelong Health and Ageing, University College London, London, UK
[2]Centre for Longitudinal Studies, Social Research Institute, University College London, London, UK
[3]Dementia Research Centre, UCL Queen Square Institute of Neurology, University College London, London, UK

**Correspondence to**
Dr Yiwen Liu; eva.liu@ucl.ac.uk

## ABSTRACT

**Objective** To investigate the accumulation of adversities (duration of exposure to any, economic, psychosocial) across the lifecourse (birth to 63 years) on cognitive function in older age, and the mediating role of mental health.

**Design** National birth cohort study.

**Setting** Great Britain.

**Participants** 5362 singleton births within marriage in England, Wales and Scotland born within 1 week of March 1946, of which 2131 completed at least 1 cognitive assessment.

**Main outcome measures** Cognitive assessments included the Addenbrooke's Cognitive Examination-III, as a measure of cognitive state, processing speed (timed-letter search task), and verbal memory (word learning task) at 69 years. Scores were standardised to the analytical sample. Mental health at 60–64 years was assessed using the 28-item General Health Questionnaire, with scores standardised to the analytical sample.

**Results** After adjusting for sex, increased duration of exposure to any adversity was associated with decreased performance on cognitive state (β=−0.39; 95% CI −0.59 to −0.20) and verbal memory (β=−0.45; 95% CI −0.63 to −0.27) at 69 years, although these effects were attenuated after adjusting for further covariates (childhood cognition and emotional problems, educational attainment). Analyses by type of adversity revealed stronger associations from economic adversity to verbal memory (β=−0.54; 95% CI −0.70 to −0.39), with a small effect remaining even after adjusting for all covariates (β=−0.18; 95% CI −0.32 to −0.03), and weaker associations from psychosocial adversity. Causal mediation analyses found that mental health mediated all associations between duration of exposure to adversity (any, economic, psychosocial) and cognitive function, with around 15% of the total effect of economic adversity on verbal memory attributable to mental health.

**Conclusions** Improving mental health among older adults has the potential to reduce cognitive impairments, as well as mitigate against some of the effect of lifecourse accumulation of adversity on cognitive performance in older age.

## STRENGTHS AND LIMITATIONS OF THIS STUDY

⇒ Used all prospective indicators of adversity spanning six decades to assess its dose–response effect on cognitive function.
⇒ Adjusted for important covariates in childhood (cognition and mental health) to mitigate against reverse directionality.
⇒ There may be bias from selective drop-out, with those who dropped out having lower childhood cognitive ability and educational attainment.
⇒ The perceived impact or timing of adversities were not considered in this study.

## INTRODUCTION

Age-related disorders account for almost a quarter of the total burden of diseases worldwide, with cognitive and mental impairments in older age being one of the leading contributors.[1] There is some evidence that exposure to adversity—often assessed using an index that captures a wide array of potentially traumatic events that may be associated with adverse functioning[2]—is associated with reduced cognitive function in older age, particularly for economic adversity such as socioeconomic circumstances (SEC), although findings remain mixed and the association between adversity and cognition is likely to be complex.[3–7] Furthermore, few studies have adopted a lifecourse approach to examine prospectively measured adversities accumulated across multiple stages of development. According to the accumulation model—a key lifecourse model—multiple (number of adversities) or persistent exposure (duration of exposure) to adversities can increase dysregulation of the hypothalamic–pituitary–adrenal (HPA)-axis, which over time could lead to physiological damage to

BMJ

neural and somatic systems.[8 9] It is therefore important to investigate whether the accumulation of adversity across the lifecourse is associated with cognitive function in older adulthood to better understand pathways to cognitive ageing.

The accumulation of adversity can be assessed as the number of adversities experienced (at the same time or across time), or the duration of exposure across time. Experiencing an increased number of adversities (often assessed at one period in time) has been associated with poorer cognitive performance and greater decline in older adulthood.[3 5 7] Similarly, studies which examined the duration of exposure to the same adversity (eg, SEC, maltreatment) also found an association between increased duration of exposure and worse cognition in mid- and older adulthood,[6 10 11] although only one study used prospectively collected measures of adversity.[11] These findings suggest that experiencing multiple adversities at the same time or experiencing the same adversity persistently across time may both be associated with cognitive impairments in later life. Further research is needed on whether experiencing any adversity (regardless of the type) persistently across the lifecourse (from childhood to older adulthood) also shows a dose–response association with poorer cognitive function in older age.

The accumulation of adversity—as well as having direct effects on cognitive function—may also have indirect effects via mental health. There is consistent evidence that both the number of and duration of exposure to adversities across the lifecourse are associated with increased psychological distress in older adulthood.[12–15] Mental ill-health was also recently identified as one of the leading modifiable risk factors for dementia[16]; both mental health and cognitive function are intertwined across the lifecourse,[17 18] and there is some evidence that mental health in mid-adulthood predicted cognition in older adulthood, but not vice versa.[17] It is therefore plausible that mental health may be a key mechanism that mediates the association between the accumulation of adversity across the lifecourse and cognitive function in older age. There is some research supporting the role of depression as a mediator between one type of adversity (ie, maltreatment, financial hardships) and cognitive function, and one study which examined the accumulation of adversity (defined as the number of adversities experienced) also identified depression as a partial mediator for cognition.[10 19] However, these studies have mainly relied on retrospective reports—which may be prone to recall biases, or did not examine the duration of exposure to adversity across different stages of the lifecourse. Therefore, further research on the mediating role of mental health is needed using prospective measures of different types of adversity across multiple stages of the lifecourse.

The aim of this study was to examine the longitudinal association between the accumulation of adversity across the lifecourse (defined as the duration of exposure to adversity) and cognitive function in older age, and the mediating role of mental health, using data from the MRC National Survey of Health and Development (NSHD; the British 1946 birth cohort). The use of this cohort is particularly advantageous given the long follow-up period, the use of repeated measures across multiple stages of the lifecourse, and the assessment of cognition and mental health in childhood. The latter is especially important for control of potential reverse directionality (ie, poorer cognitive abilities and mental health in early childhood being associated with increased exposure to adversities across the lifecourse).[20 21] We proposed two primary research questions: first, we examined the duration of exposure to any adversity from birth to 63 years with cognitive function at 69 years; second, we examined to what extent mental health at 60–64 years mediated this relationship. As secondary analyses, we examined economic and psychosocial adversity—two frequently assessed areas of adversity in the literature[22 23]—separately to test for possible specificity.

## METHODS
### Sample
The NSHD originally consisted of 5362 male and female singleton births within marriage in England, Wales and Scotland, born within 1 week of March 1946 (http://www.nshd.mrc.ac.uk/nshd). At age 69, a target sample of 2698 study members who were still alive and had a known current address in mainland Britain were invited to take part in a home visit.[24] Characteristics of those who were not assessed have been reported previously.[24] Participants provided written informed consent at each data collection. The analytical sample included in this study are those who were still alive at 69 years and completed any cognitive assessment (figure 1).

### Patient and public involvement
Patients or the public were not involved in the design, analysis or interpretation of this study.

### Measures
#### Indicators of adversity
Indicators of adversity were broadly grouped into economic and psychosocial domains across four stages of the lifecourse (see online supplemental table S1). Each was coded as binary (presence or absence).

#### *Childhood (<17 years)*
##### Economic
(1) Overcrowding (≥2 people per room); (2) lack of essential household amenities (no sole use of kitchen or bathroom, or no running hot water); (3) below average housing condition rated by the interviewer; and (4) father's occupational status (unskilled or unemployed, reported at age 11 or 15 years).

##### Psychosocial
(1) Separation from mother for ≥28 days (not due to illness or hospitalisation); (2) parental divorce; or (3)

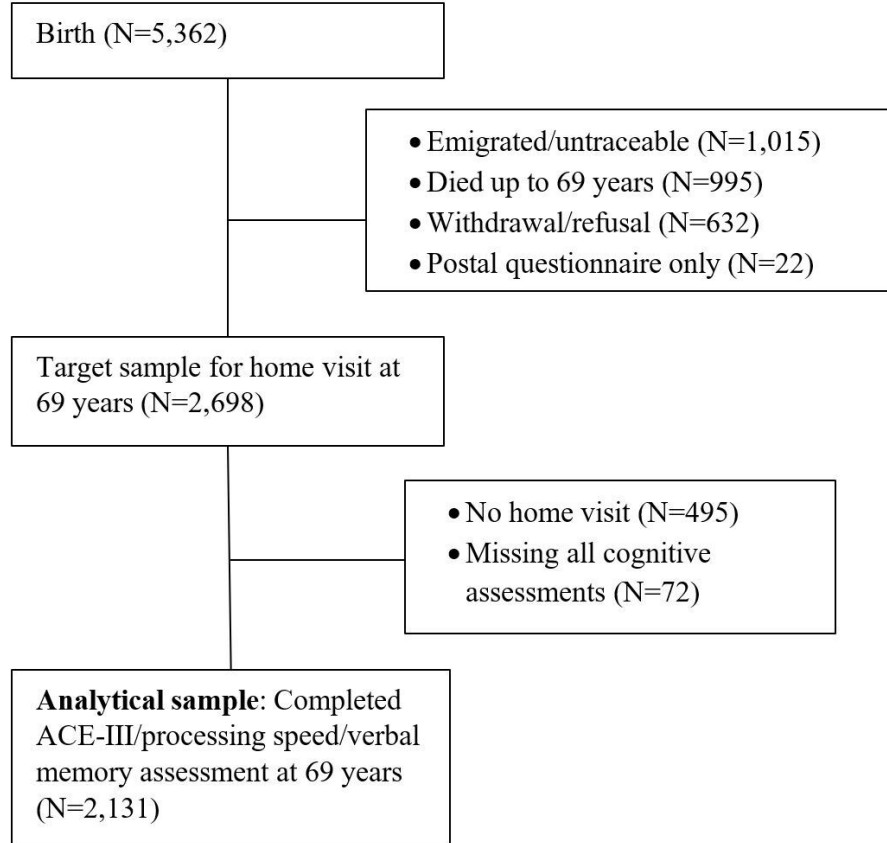

**Figure 1** Sample flow diagram. ACE-III, Addenbrooke's Cognitive Examination-III.

difficulties with peers (not able to make friends easily or not being popular with other children, rated by the teacher when participants were 13–15 years).

### Young adulthood (20–36 years)
#### Economic
(1) Overcrowding; (2) lack of essential household amenities; (3) financial hardships (going without necessities due to a shortage of money; difficulty managing on current income); or (4) unemployment.

#### Psychosocial
(1) Social isolation (never or rarely see friends or family); and (2) divorce or separation from partner.

### Mid-adulthood (43–53 years)
#### Economic
(1) Lost or feared losing employment during the last 12 months; (2) financial hardships (same items as young adulthood, with an additional item on unable to pay bills); or (3) inadequate living conditions (no central heating or running hot water, or damp in the property).

#### Psychosocial
(1) Social isolation (never see friends or family; never have people to visit); (2) lack of social support (no family or friends to talk to; have no one to help in times of crisis); (3) lost contact with relatives or friends; (4) difficulties with own children; or (5) divorce or separation from partner.

### Late adulthood (60–64 years)
The same indicators from mid-adulthood were used, apart from 'inadequate living conditions' in the economic domain, which was not assessed at 60–64 years.

#### Mental health: psychological distress (60–64 years)
Mental health at 60–64 years was assessed using the 28-item General Health Questionnaire, a self-administered questionnaire measuring psychological distress across four areas (somatic symptoms, anxiety and insomnia, social dysfunction and severe depression).[25] Each item was rated on a 4-point Likert scale from 0 (not at all) to 3 (much more than usual), with high internal reliability (α=0.91). A summary score was calculated which ranged from 0 to 84 (higher distress), and scores were standardised to the population.

### Cognitive function (69 years)
#### Cognitive state
The Addenbrooke's Cognitive Examination-III (ACE-III) was originally developed to screen for mild dementia and cognitive impairments in a clinical setting but is also an appropriate measure of general cognitive state.[26–28] It covers five domains of cognitive abilities including attention, memory, language, verbal fluency and visuospatial skills, and was conducted via the ACEmobile app on iPad, with help from the interviewer.[27 28] Scores are calculated for each cognitive domain and an overall score

(max=100) was derived from the sum of these domains, with higher scores indicating higher cognitive function.

### Processing speed
Processing speed was assessed using a timed-letter search task. Participants were instructed to cross out target letters 'P' and 'W' embedded among non-target letters as quickly and accurately as possible within 1 min. A total score representing the speed of letters identified (position reached at the end of the trial) was the main outcome, with a maximum score of 600.

### Verbal memory
Verbal memory was assessed using a recall of word learning task. Participants were shown a list of 15 words, each was presented for 2 s, and were asked to write down as many as they could remember at the end of the round. This was repeated for 3 rounds, and a total score indicating the number of words correctly recalled across all rounds was calculated, with a maximum score of 45.

### Covariates
Two sets of covariates were included: sociodemographic factors, including sex at birth (male as the reference group), and highest educational attainment by 26 years (categorised into ordinary 'O' level or below versus advanced 'A' level or above, with the latter as the reference group); and childhood measures, including cognitive abilities at 8 years (derived from 4 tests from the National Foundation for Educational Research,[29] standardised to the whole population), and emotional problems at 13–15 years, rated by the teacher and grouped into none (reference group), mild or severe.

### Statistical analyses
All statistical analyses were performed in R (V.3.6.2). Scores on each cognitive outcome were standardised to the analytical sample.

### Accumulation model
We derived a variable indicating the presence or absence of any adversity at each stage of the lifecourse (childhood, young, mid and late adulthood). These were summed to calculate the duration of exposure to any adversity and ranged from 0 (no exposure) to 4 (exposure at all four times). Using this measure, we tested for a linear dose–response association with cognitive function (cognitive state, processing speed, verbal memory) at 69 years. This was first adjusted for sex, then fully adjusted for all covariates.

### Mediation model
Linear regression models first examined the association between the duration of exposure to any adversity and psychological distress at 60–64 years, and between psychological distress at 60–64 years and cognitive functioning at 69 years, adjusted for all covariates. Causal mediation analysis then estimated the indirect effect of duration of exposure to any adversity on cognitive function via psychological distress. The R package 'mediation' was used, which performs causal mediation analysis within a counterfactual framework.[30–32] The total effect of life-course adversity on cognitive function was estimated, which is composed of an average direct effect (ADE), and an average causal mediation effect (ACME) (see online supplemental materials for detailed description on the ADE and ACME).[30–33]

### Specificity of adversity
We repeated the accumulation and mediation models for economic and psychosocial domains separately to test for possible specificity.

### Additional exploratory analysis
We further explored the interaction (tested at p<0.10 due to the increased statistical power needed to detect interaction) between economic and psychosocial adversity if both showed associations with cognitive function.

### Sensitivity analysis
We further repeated the accumulation and mediation model in a subset of participants with complete data on all three cognitive outcomes (n=1730).

### Missing data
Missing data were handled using multivariable imputation by chained equations in R[34] with 20 imputed datasets. Imputation was performed on individual indicators of adversity at each stage of the lifecourse, which were included as auxiliary variables along with all variables included in the analysis. Derived adversity variables were calculated post imputation. Imputation was performed up to the maximum analytical sample size for each cognitive outcome, and estimates were combined and pooled together using Rubin's rule.[34]

## RESULTS
### Sample characteristics
In total, 2131 participants completed at least one cognitive assessment at 69 years (cognitive state: n=1762; processing speed: n=2111; verbal memory: n=2074). The distribution of adversity variables in the sample can be found in table 1. Economic adversity was more prevalent than psychosocial adversity in childhood, young and mid-adulthood, with roughly half of participants reporting it at each stage. The rates of both adversities were reduced in older adulthood. Across the lifecourse, 4.1% of participants were not exposed to any adversity and 12.7% experienced it persistently (four times).

### Accumulation model
Increased duration of exposure to any adversity across the lifecourse was associated with lower performance on cognitive state (ACE-III) and verbal memory, but not processing speed, at 69 years. Each additional period of exposure was associated with a 0.4–0.5 SD decrease in cognitive state (β=−0.39; 95% CI −0.59 to −0.20) and

| Table 1 Sample characteristics | |
| --- | --- |
| | **n=2131** |
| **Childhood (<17 years)** | |
| Economic adversity | 1353 (63.6%) |
| Psychosocial adversity | 324 (15.2%) |
| **Young adulthood (20–36 years)** | |
| Economic adversity | 823 (40.3%) |
| Psychosocial adversity | 648 (31.4%) |
| **Mid-adulthood (43–53 years)** | |
| Economic adversity | 993 (47.4%) |
| Psychosocial adversity | 1105 (52.7%) |
| **Late adulthood (60–64 years)** | |
| Economic adversity | 316 (16.7%) |
| Psychosocial adversity | 511 (25.8%) |
| **Duration of exposure across the lifecourse to:** | |
| **Any adversity** | |
| 0 (None) | 88 (4.1%) |
| 1 (Once) | 408 (19.1%) |
| 2 (Twice) | 739 (34.7%) |
| 3 (Three times) | 626 (29.4%) |
| 4 (Four times) | 270 (12.7%) |
| **Economic** | |
| 0 (None) | 275 (12.9%) |
| 1 (Once) | 718 (33.7%) |
| 2 (Twice) | 716 (33.6%) |
| 3 (Three times) | 353 (16.6%) |
| 4 (Four times) | 69 (3.2%) |
| **Psychosocial** | |
| 0 (None) | 572 (26.8%) |
| 1 (Once) | 786 (36.9%) |
| 2 (Twice) | 543 (25.5%) |
| 3 (Three times) | 204 (9.6%) |
| 4 (Four times) | 26 (1.2%) |
| **Covariates** | |
| Sex (Female) | 1086 (51.0%) |
| Educational attainment (26 years; O-levels or below) | 1197 (59.3%) |
| **Childhood internalising problems (13–15 years)** | |
| 0 (Absent) | 1001 (52.4%) |
| 1 (Mild) | 713 (37.3%) |
| 2 (Severe) | 196 (10.3%) |
| **Mental health** | |
| Psychological distress (GHQ-28 total score; max=84) (M; SD) | 16.38 (8.02) |
| **Cognitive functioning** | |
| Cognitive state (ACE-III total score; max=100) (M; SD) | 91.52 (6.01) |
| Processing speed (Max=600) (M; SD) | 262.30 (74.14) |
| | Continued |

| Table 1 Continued | |
| --- | --- |
| | **n=2131** |
| Verbal memory (Max=45) (M; SD) | 22.17 (6.07) |

ACE-III, Addenbrooke's Cognitive Examination-III; GHQ-28, 28-item General Health Questionnaire.

verbal memory (β=−0.45; 95% CI −0.63 to −0.27) (figure 2, online supplemental table S2). However, these effect sizes were attenuated in fully adjusted models.

### Mediation model
Each increased period of exposure to any adversity across the lifecourse was associated with a 0.46 SD increase in psychological distress, and each SD increase in psychological distress was associated with a 0.07 SD decrease in cognitive state and verbal memory, and a 0.05 SD decrease in processing speed (figure 3; online supplemental table S3). Causal mediation analysis found no direct effect of duration of exposure to any adversity on cognitive functioning; however, mediated pathways were found to cognitive state and verbal memory via increased psychological distress at 60–64 years, with weaker evidence of a mediated effect found for processing speed (online supplemental table S4).

### Specificity of adversity
#### Economic
Each additional period of exposure to economic adversity was associated with a 0.51 (95% CI −0.69 to −0.32), 0.23 (95% CI −0.40 to −0.06) and 0.54 (95% CI −0.70 to −0.39) SD decrease in cognitive state, processing speed and verbal memory, respectively (figure 2; online supplemental table S2). These were attenuated in fully adjusted models, although a small effect remained on verbal memory (β=−0.18; 95% CI −0.32 to −0.03) (figure 2; online supplemental table S2). Mediated effects were found to cognitive state and verbal memory via increased psychological distress at 60–64 years, with weaker evidence for a mediated effect to processing speed (online supplemental tables S3 and S4; figure 4). Increased exposure to economic adversity showed both a direct and indirect effect on verbal memory, with around 15% of its total effect mediated via psychological distress (calculated as the ratio between total effect and mediated effect).

#### Psychosocial
A small dose–response effect was found between duration of exposure to psychosocial adversity and verbal memory (β=−0.23; 95% CI −0.46 to −0.01), although this became attenuated when adjusted for all covariates (figure 2; online supplemental table S2). No other associations were found for cognitive state or processing speed. Causal mediation analysis showed no direct effect of psychosocial adversity, although mediated effects were found via psychological distress to both cognitive state and verbal memory, with weaker evidence for mediation on processing speed (online supplemental tables S3 and S4; figure 4).

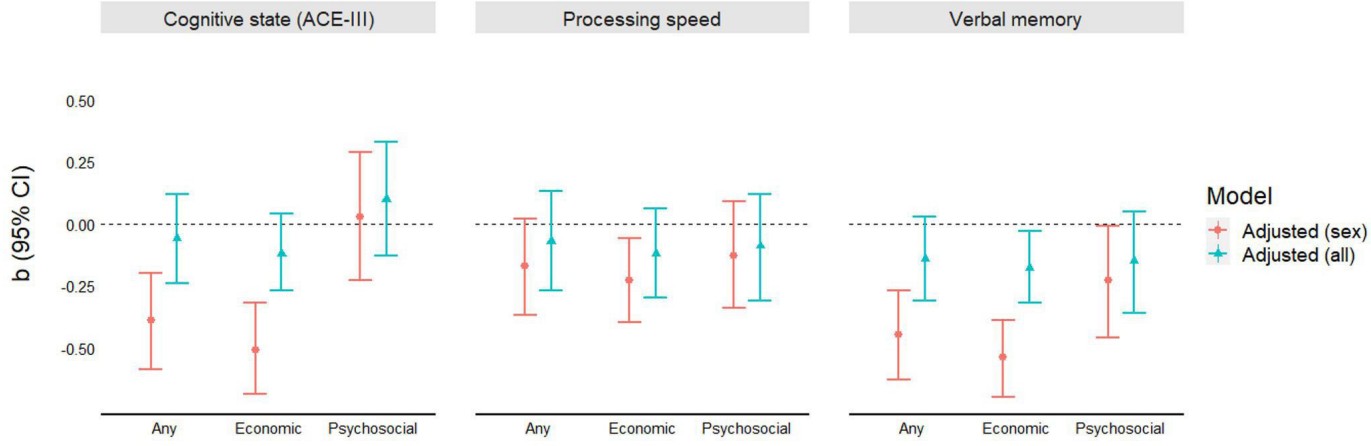

**Figure 2** Standardised estimates on the association between duration of exposure to adversity and each cognitive function at 69 years. ACE-III, Addenbrooke's Cognitive Examination-III.

### Additional exploratory analysis

Given that both economic and psychosocial adversity were associated separately with verbal memory, an interaction term between them was included in the model to examine whether there was evidence for a multiplicative effect, although no interaction was detected (p=0.597) (online supplemental table S4).

### Sensitivity analysis

Steps from the accumulation model and mediation model were repeated within a subset of participants with complete data on all three cognitive outcomes, and findings remained the same (online supplemental tables S6 and S7)

### DISCUSSION

The overall aim of this paper was to use a lifecourse approach to examine the accumulation of adversity (defined as the duration of exposure) from childhood to adulthood on cognitive function in older adults and estimate the mediating role of mental health on the causal pathway between them. Some support for the dose–response effect of the duration of exposure to adversity was found on cognitive state and verbal memory, but not processing speed. Although most of these effects were attenuated after adjusting for covariates, mediated effects via psychological distress were identified for

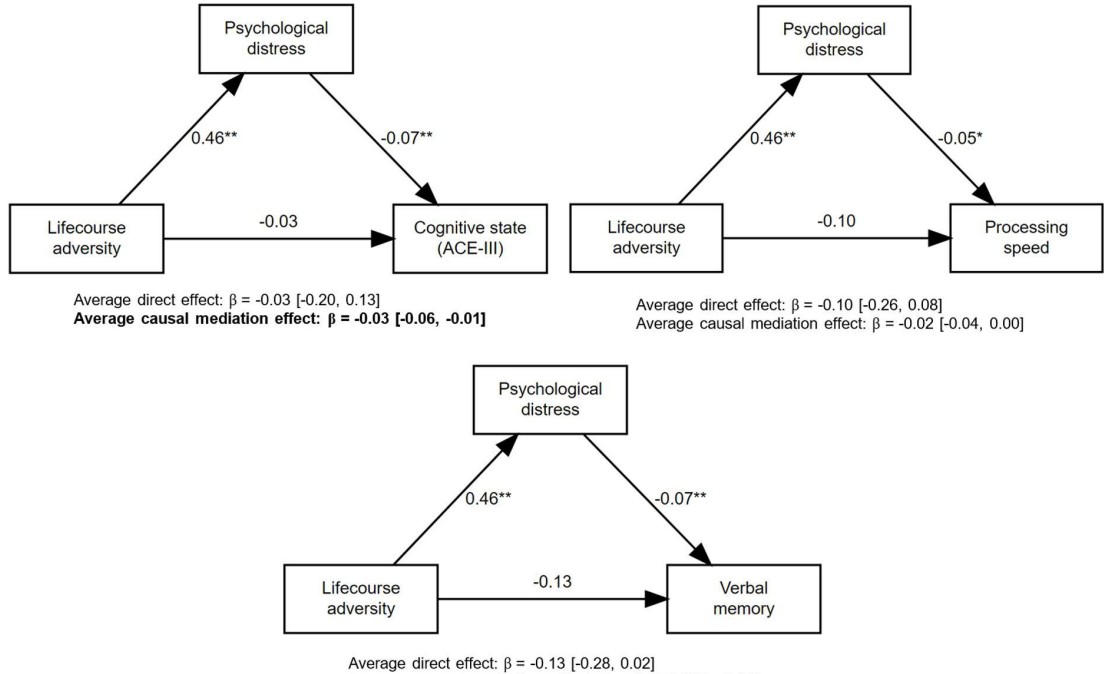

**Figure 3** Mediation models showing the direct and indirect effects of lifecourse adversity (duration of exposure to any adversity) on each cognitive function via mental health (psychological distress). *p<0.05; **p<0.01. ACE-III, Addenbrooke's Cognitive Examination-III.

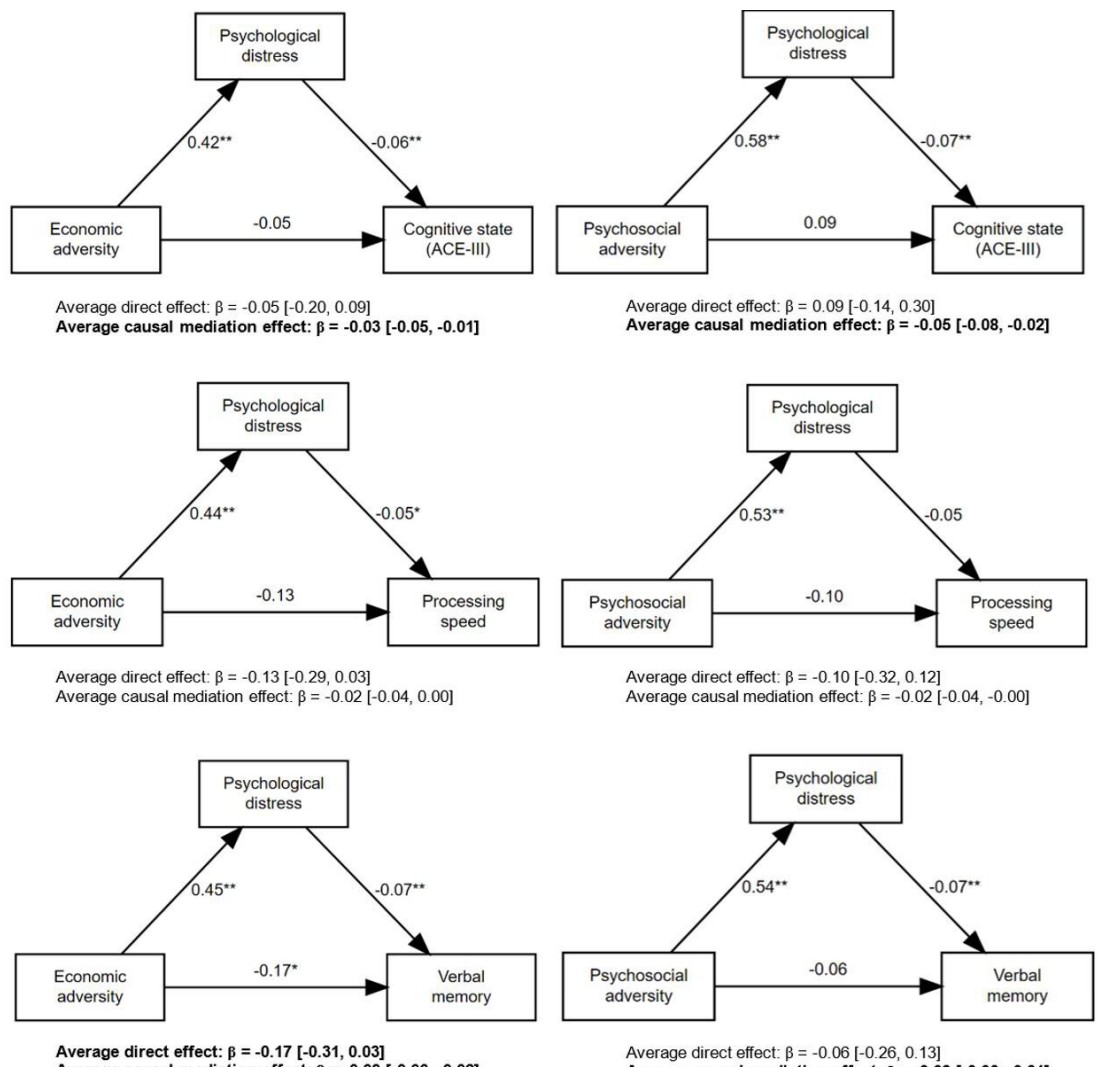

**Figure 4** Mediation models showing the direct and indirect pathways from duration of exposure to economic or psychosocial adversity to cognitive function via mental health (psychological distress). *p<0.05; **p<0.01. ACE-III, Addenbrooke's Cognitive Examination-III.

cognitive state and verbal memory. Stronger associations were found from economic compared with psychosocial adversity to all three cognitive outcomes, although similar mediated effects via psychological distress were found regardless of the type of adversity.

The accumulation of adversity can be defined using both the number of adversities experienced or the duration of exposure across time. Although the current study only focused on testing the latter, the dose–response effect of the duration of exposure to adversity on poorer cognitive function is consistent with and expands on previous research which have assessed the number of adversities, offering support for both interpretations of the accumulation model.[3 5 7 11] However, most of these associations were attenuated after adjusting for educational attainment and childhood characteristics (cognition and emotional problems),[4 35–39] highlighting the importance of having information on these outcomes from earlier in the lifecourse when trying to estimate effects in older adults. Childhood emotional problems are also

a predictor of mental ill-health in later life,[40 41] which has been associated with cognition prospectively,[17 18] and may also increase risk of being exposed to further adversities later in life.[42 43] Therefore, although increased duration of exposure to adversity may have a small dose–response effect on poorer cognitive function in older age, most of this can be accounted for by sociodemographic and cognitive and emotional function in childhood.

Although there was weaker evidence for direct effects, mediated effects were found via psychological distress to both cognitive state and verbal memory. Previous research in the same cohort have shown that the accumulation of adversity (both the number of adversities as well as the duration of exposure) is associated with increased psychological distress and decreased well-being,[15] and a few studies have also shown an indirect effect of specific types of adversity (eg, maltreatment, financial hardships) on cognition in adulthood via depressive symptoms.[10 19] However, this is the first study to our knowledge to show prospectively that increased duration of exposure to any

adversity across the lifecourse was indirectly associated with both cognitive state and specific domains of cognitive function in older adulthood via mental health in older age. One proposed mechanism is that increased psychological distress may lead to overactivation of the HPA-axis and increased cortisol production, which have been associated with cognitive impairments.[44 45] Mental ill-health in older adulthood may therefore be considered as both an antecedent for cognitive impairments later in life as well as a consequence of adversities accumulated across the lifecourse. Given that late-life depression was recently identified as one of the leading modifiable risk factors for dementia,[16] improving mental health in older adults may therefore not only directly improve cognitive function in older age, but could also mitigate some of the effect of lifecourse adversities on cognition.

Different findings also emerged when economic and psychosocial adversities were examined, with stronger associations from economic adversity to verbal memory compared with psychosocial adversity. There is some research suggesting that deprivation (eg, economic) related adversities may impact more on cognitive processes,[46] possibly via reducing cognitive bandwidth (the mental capacity for paying attention and decision-making) and increasing cognitive load,[47 48] as well as increasing brain atrophy through stress accumulation.[49 50] For example, the hippocampus in particular may be more vulnerable to economic adversity, and the accumulation of economic adversity has been previously associated with increased hippocampal atrophy in mid-to-late life.[49 50] On the other hand, psychosocial adversities may affect emotional reactivity more than cognitive functioning.[46] Although similar mediated effects via psychological distress were found from both types of adversity, this only accounted for a part of the total effect of economic adversity on verbal memory, whereas the effect of increased psychosocial adversity on poorer verbal memory was wholly mediated via poorer mental health, suggesting that mental ill-health may be the primary pathway from psychosocial adversity to worse cognitive function. Economic adversity further showed a small effect on processing speed, suggesting that processing speed may be more sensitive to economic adversity compared with other types of adversity. One potential mechanism is cardiometabolic health, which has been previously associated with both economic adversity and processing speed,[51 52] and may also mediate the association between economic adversity and cognitive function and requires further investigation.[53] Nevertheless, an important possibility is that improving mental health in older adults could help improve cognitive function in the ageing population and mitigate some of the risks associated with persistent adversities. Given the social and health costs associated with cognitive impairments and dementia[54]—combined with the cost of mental ill-health in the population[55]—increasing investment in mental health services may be crucial, in the long term, to promote healthy cognitive and mental ageing.

## Strengths and limitations

The longitudinal nature of this study offers several strengths. This is the first study to our knowledge to use prospective indicators of adversity across different lifecourse stages to show the dose-response effect of duration of exposure to adversity on poorer cognitive functioning in late adulthood. The assessment of cognition and mental health in childhood also allowed adjustment for important covariates and helped to mitigate against reverse directionality, although it does not fully exclude this possibility. The inclusion of two different domains of adversity as well as three different cognitive measures further allowed testing of specificity in the associations.

There are also some limitations that should be considered. For instance, there may be bias from selective drop-out, as previous research has suggested that those who dropped out were more likely to have lower childhood cognitive ability and educational attainment, and worse mental health.[24] This may have led to an underestimation in the association between exposure to adversity and cognitive functioning. In addition, certain adversities may not be experienced in the same way across different lifecourse stages. For example, childhood economic adversity was relatively common in this cohort, given they were born in a time of significant economic uncertainty in a post-war Britain,[56] and may be experienced differently compared with younger birth cohorts. There is some evidence suggesting that the severity of adversity should also be considered in the accumulation model, as adverse events that are perceived as life changing may show stronger effect.[9 13] Caution is also needed in assuming that the same adverse experience is being captured across the lifecourse, particularly as certain indicators of adversity (ie, difficulties with peers) was reported by the teacher. The lack of ethnic diversity in the cohort also limits the generalisability of these findings, particularly as people from ethnic minority backgrounds are more likely to experience adversities[57] and may be disproportionally affected by them. Furthermore, certain adversities may also co-occur as risk clusters or risk chains,[58–60] and adversities occurring at different stages of the lifecourse may also show differential effects. This was not investigated in the current study given our aim of testing the duration of exposure across the lifecourse. However, there is evidence for a recency effect of adversities on mental health, where later life adversities show stronger associations with mental health compared with adversities experienced earlier in the lifecourse.[13] Further research should therefore focus on assessing the severity as well as the timing of adversities, taking into account risk clusters between different adversities, and to investigate other possible mechanisms such as cardiometabolic health when examining the impact of lifecourse adversity on cognitive ageing.

## Conclusion

Overall, findings from this study suggest that increased duration of exposure to any adversity—particularly

economic adversity—has a small dose–response effect on cognitive state and verbal memory in older age. Mental health mediated the dose–response effect of duration of exposure to all types of adversity (any, economic, psychosocial) on cognition, suggesting that routinely assessing and improving mental health in older adults may also help improve cognitive function in older age. Future studies should also account for the severity, timing and co-occurrence of adversities, as well as investigate other potential mediators such as cardiometabolic health, to better understand pathways to cognitive ageing.

**Acknowledgements** We are grateful to all study members who took part in the NSHD and to the study team members who helped to collect the data over the last seven decades.

**Contributors** Conceptualisation: YL, MR and PP. Methodology: YL, PP and JS. Funding acquisition: MR, JMS and PP. Formal analysis: YL. Writing – original draft: YL. Writing – review and editing: all authors. YL serves as guarantor for the contents of this paper.

**Funding** This study is funded by the UK Medical Research Council which provides core funding for the NSHD (MC_UU_00019/1). MR additionally reports funding from MRC (MC_UU_00019/3). JMS acknowledges the support of the National Institute for Health Research University College London Hospitals Biomedical Research Centre, Alzheimer's Research UK, Medical Research Council, UK Dementia Research Institute, Alzheimer's Association, and the British Heart Foundation (PG/17/90/33415). The funders had no involvement in study design, data collection, analysis, interpretation or the decision to submit for publication. The views expressed are those of the author(s) and YL serves as guarantor for the contents of this paper.

**Competing interests** None declared.

**Patient and public involvement** Patients and/or the public were not involved in the design, or conduct, or reporting, or dissemination plans of this research.

**Patient consent for publication** Not applicable.

**Ethics approval** Ethical approval has been obtained from the Greater Manchester Local Research Ethics Committee and the Scotland Research Ethics Committee. For the latest data collection used in this study, ethical approval was obtained from the NRES Queen Square Research Ethics Committee (14/LO/1073) and Scotland A Research Ethics Committee (14/SS/1009). Participants provided written informed consent at each data collection. Participants gave informed consent to participate in the study before taking part.

**Provenance and peer review** Not commissioned; externally peer reviewed.

**Data availability statement** Data are available upon reasonable request. The data that support the findings of this study are available to bona fide researchers upon request to the NSHD Data Sharing Committee. For more information see: http://www.nshd.mrc.ac.uk/data.

**ORCID iDs**
Yiwen Liu http://orcid.org/0000-0002-2734-0182
Praveetha Patalay http://orcid.org/0000-0002-5341-3461

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
