## [Reviewer comments · BMJ Open]

ARTICLE DETAILS

TITLE (PROVISIONAL)	A lifecourse investigation of the cumulative impact of adversity on cognitive function in old age and the mediating role of mental health: longitudinal birth cohort study
AUTHORS	Liu, Yiwen; Patalay, Praveetha; Stafford, Jean; Schott, Jonathan M.; Richards, M

VERSION 1 – REVIEW

REVIEWER	Künzi, Morgane University of Oxford, Psychiatry
REVIEW RETURNED	11-Jun-2023

GENERAL COMMENTS	This study investigates the effect of accumulated lifecourse adversities on cognition in older adults and the mediating role of mental health using data from the MRC National Survey of Health and Development (NSHD). I appreciated reading this well-written manuscript that addresses well-structured and relevant research questions. My main comments, suggestions, and questions are listed below. Introduction: 1. p.4 lines 11-16: "There is some evidence that exposure to adversity is associated with reduced cognitive function in older age, particularly for economic adversity such as socio-economic circumstances (SEC)" Some studies show that the associations between adversity and cognition in older age are complex and that a negative effect of adversity on cognition is not systematically found. I think this point deserves to be discussed in the introduction.2. P.4 lines 25-30: "It is therefore important to investigate whether persistent exposure to adversity across the lifecourse is associated with cognitive function in older adulthood to promote healthy cognitive ageing." It would be good to define the term persistent in accordance with lines 42-49 (i.e., multiple adversities throughout life, adversity that lasts over time, or both).3. On several occasions in the introduction, the use of prospective measures of adversity as opposed to retrospective measures (which are often used) is recommended. However, an explanation of why the use of prospective measures is beneficial over retrospective measures is lacking.4. P.5 lines 39-43: The advantage of the use of this dataset for its long follow-up was highlighted. Therefore, I wonder why the
---

	authors did not also use longitudinal data for mental health and cognition, which would have been interesting to study. 5. A definition of adversity and the experiences it encompasses is lacking. Adding a definition would make the choice of certain adversities clearer later on. Methods:  1. P.7 line 23 "childhood (< 16 years)", in the supplementary material it is written <17. 2. Throughout the manuscript, the use of prospective measures of adversity is highlighted as an advantage of this study, why then do the authors use retrospective measures (e.g., affectionless control parenting, maltreatment) which may thus be subject to various biases (e.g., mental health)? 3. I am concerned about the choice of certain adversities and their grouping.  a. For example, social isolation may not be associated with the feeling of loneliness, and therefore it could be that social isolation is a choice rather than an adversity. b. I would not have classified "any work-related stress" as economic adversity, I would like to hear the authors' views on the rationale for this choice. c. There are differences within the same life-stage category in terms of adversity severity (i.e., maltreatment and ability to make friends). Similarly, there are also differences between life stages within the same category (i.e., maltreatment and separation/divorce from partner, or social isolation). How did the authors take these differences into account? 4. P.11 lines 24-27, in the additional exploratory analysis, it would be good to justify why the interaction is tested at $p < .10$. Results:  1. Table 1. Sample characteristics, it is written "childhood (< 17 years)" contrary to the methods. 2. The Fig. S1 is very clear, it could be helpful to have this figure in the manuscript. 3. Table S3. There is a typing error in the word association and two dots in the table footnotes. Discussion:  1. In the manuscript, it is sometimes written "duration of adversity", "accumulation of adversity" and "accumulating effect of duration", which creates confusion between number and duration and makes the manuscript less straightforward. I suggest that the authors clarify this point throughout the manuscript to avoid any confusion. 2. An important point in the manuscript is the cumulative aspect of adversity and the aspect of duration, yet this point is addressed very little in the introduction and not at all in the discussion. I would like the authors to discuss this important point in the manuscript. 3. No significant association between lifecourse adversity and processing speed was found compared to lifecourse adversity and
--	---

	cognitive state as well as verbal memory, I would find it interesting to discuss this. 4. P.17 line 21, not all the adversities are prospective, I would be more cautious and either put less emphasis on the use of prospective adversity measures or remove from the analyses the adversity retrospectively reported. 5. P.18 lines 19-24, I would replace “chronic adversities” with long-lasting adversities because “chronic adversities” is a new term that has not been introduced before and may add confusion with the terms used previously (i.e., accumulation and duration). 6. Regarding the drop-out, there may also be selection bias with regard to cognition and mental health: those who remain in the study may be in better cognitive and mental health. 7. Other undiscussed biases could be present, such as the cohort effect, a third party reporting adversity (i.e., teacher) but also mental health influencing the self-report of retrospective adversity. How might these biases affect your results?
--	---

REVIEWER	Halpin, Amy The University of Maine
REVIEW RETURNED	16-Jun-2023

GENERAL COMMENTS	Thank you for the opportunity to review the article “A lifecourse investigation of the cumulative impact of adversity on cognitive function in old age and the mediating role of mental health: longitudinal birth cohort study.” This study provides an interesting and needed expansion to our understanding of how economic and psychosocial adversity can impact late life cognitive functioning. The authors address several important gaps in the literature: a) the longitudinal impacts of adverse experiences on cognitive functioning, b) examining the directionality of the relationships between mental health, adversity, and cognitive functioning, and c) understanding if a dose-dependent relationship exists between duration and severity of adversity with cognitive functioning. The authors provided sound rationale for their study and identified important gaps in the research in setting up their objectives. Their use of a large cohort served as a strength of their study and increased confidence in findings. The authors further used a comprehensive cognitive screener to ascertain cognitive status. Most compelling of all was the use of longitudinal data and a sophisticated analysis to allow for directionality of findings. authors also provided a relevant and appropriate discussion to support their findings. Despite this, the article would benefit from additional considerations. Specific comments follow. Introduction 1. Line 45 states: “early cognitive development and mental health being associated with exposure to adversities across the lifecourse”---What specifically about early cognitive development and mental health is associated with exposure to adversity? Is it slower cognitive development and worse mental health? Is it delayed cognitive development? As it is written now, it is too vague. 2. Why was the age of 63 used to ascertain mental health status? Perhaps providing an explanation as to the significance of these
---

	two ages or the significance of the duration between them would be appreciated by the reader. Methods  1. The retrospective report used to define the psychosocial adversity for the childhood period should be listed as a limitation given reliability issues commonly associated with retrospective report. 2. Was the reliability of the General Health Questionnaire conducted for the present sample? Similarly, are there cut points or clinical thresholds described elsewhere to provide a basis for what scores are considered “normal” vs “abnormal?” 3. What specific verbal memory measure was used? It is not clear from the methods description. Results  1. It would be interesting to understand if the accumulation model was impacted at all by the time of the adversity (e.g., is adversity experienced later in life more salient to later mental life and/or cognitive functioning?) 2. It does not appear that the cognitive outcome variable was the same across all 2,131 participants. This makes it challenging to derive conclusions about how life course adversity impacted later life cognition, as different participants were associated with the different outcome variables. 3. It would be helpful to the reader to have the direct and indirect effects of lifecourse adversity on each of the cognitive outcomes listed on the mediation models themselves (figure 3) Discussion  1. It is recommended that the authors provide more specific directions for future research in the discussion. 2. More of a discussion on the connections between early economic deprivation and cognitive processes would be appreciated since this was a finding in their study. They mention it may have an impact via “reduced cognitive bandwidth and increasing cognitive load as well as increasing brain atrophy through stress accumulation”, but more of a discussion is warranted. 3. It is suggested that the authors add a section to consider how recent adversity may be felt more strongly than early adversity, and therefore may have more of an impact on late life mental health and/or late life cognitive functioning.
--	---

VERSION 1 – AUTHOR RESPONSE

Reviewer: 1 Dr. Morgane Künzi, University of Oxford Comments to the Author: This study investigates the effect of accumulated lifecourse adversities on cognition in older adults and the mediating role of mental health using data	Thank you very much for your positive feedback, we are pleased to have addressed your comments below.	
---	--	--

from the MRC National Survey of Health and Development (NSHD). I appreciated reading this well-written manuscript that addresses well-structured and relevant research questions. My main comments, suggestions, and questions are listed below.		
Introduction: 1. p.4 lines 11-16: "There is some evidence that exposure to adversity is associated with reduced cognitive function in older age, particularly for economic adversity such as socio-economic circumstances (SEC)" Some studies show that the associations between adversity and cognition in older age are complex and that a negative effect of adversity on cognition is not systematically found. I think this point deserves to be discussed in the introduction.	We have now clarified in the introduction that findings on SEC and cognition remain mixed and the association between them is likely to be complex.	4
2. P.4 lines 25-30: "It is therefore important to investigate whether persistent exposure to adversity across the lifecourse is associated with cognitive function in older adulthood to promote healthy cognitive ageing." It would be good to define the term persistent in accordance with lines 42-49 (i.e., multiple adversities throughout life, adversity that lasts over time, or both).	Thank you, we have now defined the term persistent (duration of exposure to adversities) in the introduction paragraph.	4
3. On several occasions in the introduction, the use of prospective measures of adversity as opposed to retrospective measures (which are often used) is recommended. However, an explanation of why the use of prospective measures is beneficial over retrospective measures is lacking.	Thank you for raising this point, we have now added the limitation of using retrospective measures such as recall bias.	5

4. P.5 lines 39-43: The advantage of the use of this dataset for its long follow-up was highlighted. Therefore, I wonder why the authors did not also use longitudinal data for mental health and cognition, which would have been interesting to study.	Thank you for your comment. We used longitudinal data for adversity as we were interested in the duration of exposure to adversity across the lifecourse, rather than longitudinal changes in mental health or cognition (which would require different models such as a cross-lagged model). We agree it is interesting to investigate changes in cognition over time, and this is the topic of another paper we are working on. We chose mental health at 63 years (it is actually 60-64 years, as data collection took more than one year during this wave) due to the testing of mental health as a mediator.	
5. A definition of adversity and the experiences it encompasses is lacking. Adding a definition would make the choice of certain adversities clearer later on.	Thank you for your suggestion, we have now included this in introduction, where adversity is defined as any potentially traumatic event that may be associated with adverse functioning, and often assessed using an index in previous literature.	4
Methods: 1. P.7 line 23 "childhood (< 16 years)", in the supplementary material it is written <17.	Thank you for pointing this out, we have changed it to <17 years as it is assessed up to and including 16 years.	8 (7 clean version)
2. Throughout the manuscript, the use of prospective measures of adversity is highlighted as an advantage of this study, why then do the authors use retrospective measures (e.g., affectionless control parenting, maltreatment) which may thus be subject to various biases (e.g., mental health)?	Thank you for raising this point, we have now decided to remove these retrospective measures in order to strengthen the paper by using only prospectively measured adversities. Findings remained similar after removing these measures.	8
3. I am concerned about the choice of certain adversities and their grouping. a. For example, social isolation may not be associated with the feeling of loneliness, and therefore it could be that social isolation is a choice rather than an adversity. b. I would not have classified "any work-related stress" as economic adversity, I would like to hear the authors' views on the rationale for this choice. c. There are differences within the same life-stage category in terms of adversity severity (i.e.,	Thank you for raising this issue, we agree that the groupings are not always ideal and that adversities within the same domain do not always reflect the same type of experience. a) We included social isolation (which is assessed as the frequency of contact with friends/family) as there is evidence that infrequent social contact is an important risk factor for dementia (Livingston et al., 2020: https://doi.org/10.1016/S0140-6736(20)30367-6). However, we also recognise that different people will experience social isolation in different ways. Although we do have some information on participants' experience of certain adversities (e.g., participants were asked whether an event was life changing), these were not routinely assessed, and we did not want to exclude variables based on this. We have now discussed this as a limitation in the discussion section of the paper, and recommend future studies to also assess the severity or perceived impact of different adverse events.	8, 23 (22 clean version)

maltreatment and ability to make friends). Similarly, there are also differences between life stages within the same category (i.e., maltreatment and separation/divorce from partner, or social isolation). How did the authors take these differences into account?	b) We further agree that “any work-related stress” is a vague term and may not represent economic adversity. This item originally consisted of two measures: “having lost or feared losing employment in the last 12 months”, and “any other crisis experienced at work”. We have decided to only use the first measure (lost/fearful losing employment) as we feel this is more representative of economic adversity as it relates to employment instability, and removed the second measure (crisis at work) from all models. Findings remained similar after removing this, with a stronger association found between economic adversity and processing speed, suggesting the remaining indicators of economic adversity were more relevant for this cognitive domain. c) Lastly, thank you for raising the issue of the severity of adversities such as maltreatment in comparison to other psychosocial indicators like divorce and social isolation. We have now removed maltreatment from all models, as although it is an important indicator of childhood adversity, unfortunately this was only assessed retrospectively using one item, therefore in order to strengthen the paper by keeping all indicators of adversity prospective, we have decided to remove this. The issue with the severity or perceived impact of other adversities have also been discussed in the limitation section of the paper.	
4. P.11 lines 24-27, in the additional exploratory analysis, it would be good to justify why the interaction is tested at $p < .10$.	We chose a larger p value in order to capture any interactions that might not have reached traditional significance level ($p < 0.05$) due to the higher statistical power needed to detect interactions, which is the standard convention in epidemiology.	12 (11 clean version)
Results: 1. Table 1. Sample characteristics, it is written “childhood (< 17 years)” contrary to the methods.	Thank you for spotting this typo, we have now amended the methods section to <17 years.	8 (7 clean version)
2. The Fig. S1 is very clear, it could be helpful to have this figure in the manuscript.	Thank you, we have now included this in the main manuscript.	Fig 4
3. Table S3. There is a typing error in the word association and two dots in the table footnotes.	Thank you for pointing this out, we have corrected these.	Supplements
Discussion: 1. In the manuscript, it is sometimes written “duration of adversity”, “accumulation of adversity” and “accumulating effect of duration”, which creates confusion between number and duration and	Thank you for pointing out the inconsistencies in the terms used. We now use the term “accumulation of adversity” in the introduction and explained that it can be defined as the number of adversities or the duration of exposure. Subsequently, whenever we refer to the accumulation of adversity, we have clarified which definition we are referring to, and stated in our aims that we will only be focusing on the duration of adversity (as this has been less examined in the literature compared to the number of	4-6, throughout paper

makes the manuscript less straightforward. I suggest that the authors clarify this point throughout the manuscript to avoid any confusion.	adversities). We have also removed the term “accumulating effect of duration”.	
2. An important point in the manuscript is the cumulative aspect of adversity and the aspect of duration, yet this point is addressed very little in the introduction and not at all in the discussion. I would like the authors to discuss this important point in the manuscript.	Thank you for your suggestion. We have now included further discussion on the differences between the two definitions of the accumulation of adversity (number of adversities and duration of exposure). We have also included a brief discussion of the differences between them in the discussion, as the aim of the paper was on the duration of exposure rather than number.	4, 5, 19, 20
3. No significant association between lifecourse adversity and processing speed was found compared to lifecourse adversity and cognitive state as well as verbal memory, I would find it interesting to discuss this.	Thank you for pointing this out. After removing “any other crisis at work” from the economic adversity domain, we now find an association between economic adversity and processing speed (before adjusting for covariates), suggesting that the remaining measures of economic adversity were more relevant and sensitive in capturing differences in processing speed compared to the one we removed (i.e., any other crisis at work). Another explanation for the smaller effects found for processing speed could be due to other mechanisms such as cardiometabolic health, which has been previously associated with both economic adversity and processing speed. It is plausible that cardiometabolic health may be an alternative mechanism in which lifecourse adversity (and specifically economic adversity) is associated with processing speed, and this is now discussed in the discussion section of the paper.	21, 22 (20 clean version)
4. P.17 line 21, not all the adversities are prospective, I would be more cautious and either put less emphasis on the use of prospective adversity measures or remove from the analyses the adversity retrospectively reported.	This is a good point and we have now removed retrospective measures of adversity from the analysis.	8
5. P.18 lines 19-24, I would replace “chronic adversities” with long-lasting adversities because “chronic adversities” is a new term that has not been introduced before and may add confusion with the terms used previously (i.e., accumulation and duration).	Thank you for raising this and we have now replaced this with “persistent adversities” as we have previously defined this.	22 (21 clean version)

6. Regarding the drop-out, there may also be selection bias with regard to cognition and mental health: those who remain in the study may be in better cognitive and mental health.	Thank you for your suggestion, this is an important point and we have added this to the limitations section.	22 (21 clean version)
7. Other undiscussed biases could be present, such as the cohort effect, a third party reporting adversity (i.e., teacher) but also mental health influencing the self-report of retrospective adversity. How might these biases affect your results?	Thank you for pointing out these importance sources of bias, we have now included a more detailed discussion on these biases in the limitations section.	23 (21-22 clean version)
Reviewer: 2 Dr. Amy Halpin, The University of Maine Comments to the Author: Thank you for the opportunity to review the article “A lifecourse investigation of the cumulative impact of adversity on cognitive function in old age and the mediating role of mental health: longitudinal birth cohort study.” This study provides an interesting and needed expansion to our understanding of how economic and psychosocial adversity can impact late life cognitive functioning. The authors address several important gaps in the literature: a) the longitudinal impacts of adverse experiences on cognitive functioning, b) examining the directionality of the relationships between mental health, adversity, and cognitive functioning, and c) understanding if a dose-dependent relationship exists between duration and severity of adversity with cognitive functioning. The authors provided sound rationale for their study and identified important gaps in the research in setting up their objectives.	Thank you very much for your positive feedback and we are pleased to have addressed your comments below.	

Their use of a large cohort served as a strength of their study and increased confidence in findings. The authors further used a comprehensive cognitive screener to ascertain cognitive status. Most compelling of all was the use of longitudinal data and a sophisticated analysis to allow for directionality of findings. authors also provided a relevant and appropriate discussion to support their findings. Despite this, the article would benefit from additional considerations. Specific comments follow.		
Introduction 1. Line 45 states: “early cognitive development and mental health being associated with exposure to adversities across the lifecourse”---What specifically about early cognitive development and mental health is associated with exposure to adversity? Is it slower cognitive development and worse mental health? Is it delayed cognitive development? As it is written now, it is too vague.	Thank you for bringing this to our attention, we have now explained that it is poorer cognitive abilities and poorer mental health in childhood being associated with increased exposure to adversities.	6
2. Why was the age of 63 used to ascertain mental health status? Perhaps providing an explanation as to the significance of these two ages or the significance of the duration between them would be appreciated by the reader.	We selected mental health at 63 years (corrected to 60-64 years due to data collection taking more than one year during this wave) because mental health was considered as a mediator in this study.	
Methods 1. The retrospective report used to define the psychosocial adversity for the childhood period should be listed as a limitation given reliability issues commonly associated with retrospective report.	Thank you for raising this valid point and we have now removed these retrospective measures from our analysis. Findings remained similar after removing these.	8

2. Was the reliability of the General Health Questionnaire conducted for the present sample? Similarly, are there cut points or clinical thresholds described elsewhere to provide a basis for what scores are considered “normal” vs “abnormal?”	Thank you for your question, there is high internal reliability for the GHQ in the current sample ($\alpha = 0.91$) which we have now included in the paper. There are traditional cut off points for using the GHQ as a screening test (e.g., Goldberg & Hillier [https://doi.org/10.1017/s0033291700021644] recommended dichotomising at each item level, with >1 as the cut-off, summing these and having a cut-off of 4/5). However, in the present study we have used the GHQ as a continuous score, as it allows us to capture more variation than dichotomising it.	9
3. What specific verbal memory measure was used? It is not clear from the methods description.	Thank you for your question, verbal memory was assessed using a recall of a 15-item word learning task. Each word was presented for 2 seconds, and participants were asked at the end of the round to write down as many as they could remember. This was repeated over 3 rounds and the total score represent the number of words correctly recalled across all rounds (max = 45). We have added further details in the methods section.	10
Results 1. It would be interesting to understand if the accumulation model was impacted at all by the time of the adversity (e.g., is adversity experienced later in life more salient to later mental life and/or cognitive functioning?)	Thank you for raising this important point regarding the timing of adversity. We agree it is also important to consider the timing of exposure (such as testing the sensitive period model or recency effect). However, the current paper is primarily interested in testing the duration of exposure across the lifecourse (rather than the number of adversities experienced at a particular stage of the lifecourse), therefore we did not examine the timing model in the current paper as it falls outside our aims. However, we have now included a discussion on the recency effect of adversity on mental health outcomes, and added this as a possible future direction to examine the timing of adversity in relation to cognitive outcomes.	23 (22 clean version)
2. It does not appear that the cognitive outcome variable was the same across all 2,131 participants. This makes it challenging to derive conclusions about how life course adversity impacted later life cognition, as different participants were associated with the different outcome variables.	Thank you for raising this point, this is indeed an important issue to consider, and we have now performed a sensitivity analysis by only including participants who have completed assessments on all three cognitive outcomes (N=1720). Findings remained similar compared to the main analysis, which provides us with more confidence that although different samples were used, this did not affect the validity of our results.	12, 19 (18 clean version), supplements
3. It would be helpful to the reader to have the direct and indirect effects of lifecourse adversity on each of the cognitive outcomes listed on the mediation models themselves (figure 3)	Thank you for your suggestion, we agree this would be clearer and have now added the direct and mediated effects for each model to the figures.	Fig 3, Fig 4
Discussion 1. It is recommended that the authors provide more specific directions for	Thank you for your comment, we have now added more details on directions for future research, including (1) assessing the severity of adversity (such as whether adversities were considered as life	23, 24 (22, 23 clean version)

future research in the discussion.	changing); (2) consider the timing of exposure; (3) examine risk chains between different adversities; and (4) investigate other mechanisms as cardiometabolic health.	
2. More of a discussion on the connections between early economic deprivation and cognitive processes would be appreciated since this was a finding in their study. They mention it may have an impact via “reduced cognitive bandwidth and increasing cognitive load as well as increasing brain atrophy through stress accumulation”, but more of a discussion is warranted.	Thank you for your comment, we have now expanded this further, with a definition of what we mean by cognitive bandwidth, as well as including specific examples of how stress accumulation affects brain atrophy (economic adversity being shown to accelerate hippocampal atrophy from previous research).	21 (20 clean version)
3. It is suggested that the authors add a section to consider how recent adversity may be felt more strongly than early adversity, and therefore may have more of an impact on late life mental health and/or late life cognitive functioning.	Thank you for your suggestion, we have included this in the discussion section, where we acknowledge that the timing of adversities may also have differential effects on mental health and cognition. However, due to the focus of the paper being on the duration of exposure across the lifecourse (rather than the accumulation of adversity within specific stages of the lifecourse), we did not examine the timing models (sensitive period/recency effect), but have included this as an important step for future work.	23, 24 (22 clean version)

VERSION 2 – REVIEW

REVIEWER	Künzi, Morgane University of Oxford, Psychiatry
REVIEW RETURNED	18-Aug-2023

GENERAL COMMENTS	Thank you for addressing the reviewers' comments. The proposed revision is comprehensive and addresses the questions and points raised by the reviewers.
--

REVIEWER	Halpin, Amy The University of Maine
REVIEW RETURNED	16-Aug-2023

GENERAL COMMENTS	Thank you for your thoughtful consideration and approach to addressing concerns expressed by both reviewers and the editor. Your revision to the manuscript is much improved; however, I would like to see additional attention given to increasing detail and specificity throughout the manuscript. I have identified specific examples in the attached document: Thank you to the authors for their thoughtful consideration and approach to addressing concerns expressed by both reviewers and the editor. The authors' revision to the manuscript is much improved; however, I would like to see additional attention given toward increasing detail and specificity throughout the manuscript
--

	by using appropriate comparative terms (e.g., worse, better, poor, etc.) when discussing relationships between adversity and mental health/cognitive functioning. I have provided some examples below: Introduction  1. The last sentence of the first paragraph, “It is therefore important to investigate whether the accumulation of adversity across the lifecourse is associated with cognitive function in older adulthood to promote healthy cognitive ageing”, could be improved for clarity. Right now, it reads as if finding a link between adversity and cognitive functioning will promote healthy aging. 2. The sentence of the second paragraph needs further directionality. Do the authors mean a dose-response association with worse cognitive function in older age? 3. The third sentence in the third paragraph could use revision. “Mental health” is not a risk factor; however, poor mental health can be. Discussion  1. Page 16, the sentence “Childhood emotional problems are also a predictor of mental health in later life..”, should be further clarified. I am assuming the authors mean, worse mental health? Or poor mental health? 2. The sentence, “...whereas the effect of psychosocial adversity on verbal memory was wholly mediated, suggesting that mental health may be the primary pathway from psychosocial adversity to cognitive function” could also benefit from more specificity by adding comparative descriptors (e.g., worse, poor, better, enhanced, etc.).
--	---

VERSION 2 – AUTHOR RESPONSE

Many thanks to the reviewer for their further comments on improving the manuscript. Please see our response and changes made in red below (and marked as tracked changes in the revised manuscript).

Thank you to the authors for their thoughtful consideration and approach to addressing concerns expressed by both reviewers and the editor. The authors’ revision to the manuscript is much improved; however, I would like to see additional attention given toward increasing detail and specificity throughout the manuscript by using appropriate comparative terms (e.g., worse, better, poor, etc.) when discussing relationships between adversity and mental health/cognitive functioning. I have provided some examples below:

Introduction

1. The last sentence of the first paragraph, “It is therefore important to investigate whether

the accumulation of adversity across the lifecourse is associated with cognitive function in older adulthood to promote healthy cognitive ageing”, could be improved for clarity. Right now, it reads as if finding a link between adversity and cognitive functioning will promote healthy aging.

Thank you for pointing this out, we have replaced “to promote healthy cognitive ageing” with “to better understand pathways to cognitive ageing” to avoid confusion (pg 4).

2. The sentence of the second paragraph needs further directionality. Do the authors mean a dose-response association with worse cognitive function in older age?

Thank you, we have now clarified that it is associated with worse/poorer cognitive function (pg 4-5).

3. The third sentence in the third paragraph could use revision. “Mental health” is not a risk factor; however, poor mental health can be.

Thank you, we have now clarified it is mental ill-health, rather than mental health, that is a risk factor for dementia (pg 5).

Discussion

1. Page 16, the sentence “Childhood emotional problems are also a predictor of mental health in later life..”, should be further clarified. I am assuming the authors mean, worse mental health? Or poor mental health?

Clarified that it is mental ill-health, thank you for pointing this out (pg 16).

2. The sentence, “...whereas the effect of psychosocial adversity on verbal memory was wholly mediated, suggesting that mental health may be the primary pathway from psychosocial adversity to cognitive function” could also benefit from more specificity by adding comparative descriptors (e.g., worse, poor, better, enhanced, etc.).

Thank you, we have expanded this to “the effect of increased psychosocial adversity on poorer verbal memory was wholly mediated via poorer mental health, suggesting that mental ill-health may be the primary pathway from psychosocial adversity to worse cognitive function” (pg 17).

We also checked the manuscript and added comparative terms as appropriate throughout.